# Pancreatic Cancer Surveillance in Carriers of a Germline Pathogenic Variant in *CDKN2A*

**DOI:** 10.3390/cancers15061690

**Published:** 2023-03-09

**Authors:** Joan Llach, Paula Aguilera, Ariadna Sánchez, Angels Ginès, Glòria Fernández-Esparrach, Guillem Soy, Oriol Sendino, Eva Vaquero, Sabela Carballal, Fabio Ausania, Juan Ramón Ayuso, Anna Darnell, María Pellisé, Sergi Castellví-Bel, Susana Puig, Francesc Balaguer, Leticia Moreira

**Affiliations:** 1Department of Gastroenterology, Hospital Clínic Barcelona, University of Barcelona, 08036 Barcelona, Spain; jllachr@clinic.cat (J.L.); asanchezg@clinic.cat (A.S.); magines@clinic.cat (A.G.); mgfernan@clinic.cat (G.F.-E.); gsoy@clinic.cat (G.S.); sendino@clinic.cat (O.S.); evaquero@clinic.cat (E.V.); carballal@clinic.cat (S.C.); mpellise@clinic.cat (M.P.); fprunes@clinic.cat (F.B.); 2Centro de Investigación Biomédica en Red en Enfermedades Hepáticas y Digestivas (CIBEREHD), 08036 Barcelona, Spain; sbel@recerca.clinic.cat; 3IDIBAPS (Institut d’Investigacions Biomèdiques August Pi i Sunyer), University of Barcelona (UB), 08036 Barcelona, Spain; ausania@clinic.cat (F.A.); jrayuso@clinic.cat (J.R.A.); andarnel@clinic.cat (A.D.); 4Dermatology Department, Hospital Clínic Barcelona, University of Barcelona, 08036 Barcelona, Spain; paguile@clinic.cat (P.A.); spuig@clinic.cat (S.P.); 5Centro de Investigación Biomédica en Red de Enfermedades Raras Instituto de Salud Carlos III, 08036 Barcelona, Spain; 6Facultat de Medicina I Ciències de la Salut, Universitat de Barcelona (UB), 08036 Barcelona, Spain; 7Department of General and Digestive Surgery, Hospital Clínic Barcelona, University of Barcelona, 08036 Barcelona, Spain; 8Department of Radiology, Hospital Clínic Barcelona, University of Barcelona, 08036 Barcelona, Spain

**Keywords:** pancreatic cancer, surveillance, hereditary, *CDKN2A*

## Abstract

**Simple Summary:**

Pancreatic cancer surveillance in high-risk individuals is not well-defined, and recommendations in this regard have recently changed. Specifically, in carriers of a germline pathogenic variant in *CDKN2A*, the risk of pancreatic ductal adenocarcinoma seems to be higher than previously reported. We present a cohort with a large number of *CDKN2A* heterozygotes under surveillance, in which an early pancreatic cancer was detected and a curative treatment was offered. Our data support the latest surveillance recommendations in these individuals.

**Abstract:**

Three percent of patients with pancreatic ductal adenocarcinoma (PDAC) present a germline pathogenic variant (GPV) associated with an increased risk of this tumor, *CDKN2A* being one of the genes associated with the highest risk. There is no clear consensus on the recommendations for surveillance in *CDKN2A* GPV carriers, although the latest guidelines from the International Cancer of the Pancreas Screening Consortium recommend annual endoscopic ultrasound (EUS) or magnetic resonance imaging (MRI) regardless of family history. Our aim is to describe the findings of the PDAC surveillance program in a cohort of healthy *CDKN2A* GPV heterozygotes. This is an observational analysis of prospectively collected data from all *CDKN2A* carriers who underwent screening for PDAC at the high-risk digestive cancer clinic of the “Hospital Clínic de Barcelona” between 2013 and 2021. A total of 78 subjects were included. EUS or MRI was performed annually with a median follow-up of 66 months. Up to 17 pancreatic findings were described in 16 (20.5%) individuals under surveillance, although most of them were benign. No significant precursor lesions were identified, but an early PDAC was detected and treated. While better preventive strategies are developed, we believe that annual surveillance with EUS and/or MRI in *CDKN2A* GPV heterozygotes may be beneficial.

## 1. Introduction

Pancreatic ductal adenocarcinoma (PDAC) is a rapidly progressive disease that causes more than 466,000 deaths per year. It is the seventh leading cause of cancer death in both sexes worldwide and, it is estimated that by 2030, it will be the second leading cause of death from cancer in the USA [1,2].

Family history of PDAC has been associated with an increased risk of developing this tumor. Approximately 5–10% of patients with PDAC have relatives with this neoplasm, which indicates the existence of familial risk factors in the pathogenesis of this disease [3,4,5]. There are two clinical situations in which a familial predisposition to PDAC has been described: familial PDAC (FPDAC), in which a familial aggregation of PDAC is observed with no identified hereditary cause [5] and hereditary PDAC, in which there is an association with a germline pathogenic variant (GPV) that carries an increased risk of developing this tumor. These hereditary cancer syndromes represent 3% of all PDACs but, except for hereditary pancreatitis, these GPVs also predispose the patients to other tumors, and the pancreas is not the main organ affected. The inherited syndromes associated with an increased risk of PDAC and specific GPV are Peutz–Jeghers syndrome [6], familial atypical multiple mole melanoma (FAMMM) [7], hereditary breast–ovarian cancer syndrome [5,8], familial adenomatous polyposis syndrome [9], Lynch syndrome [10], Li–Fraumeni syndrome, and ataxia telangiectasia [11,12].

Of these syndromes, FAMMM, which is characterized by a high risk of developing multiple dysplastic nevi and melanoma with a cumulative risk of 60–80% at 80 years, is also associated with a high risk of developing PDAC. Its cumulative lifetime risk is up to 20% with a 13–22-fold increased risk compared to the general population [13]. *CDKN2A* is a dominant inherited gene located on chromosome 9p21 [14] that encodes two proteins: p16 and p14ARF, the first protein being a negative regulator of cell cycle progression [15]. In this syndrome, the increased risk of PDAC has been associated especially with the pathogenic variant of p16 [7].

Although treatments for PDAC are improving, the 5 year survival rate is still less than 5%, and the main reason is its diagnosis in advanced stages. Therefore, those individuals with significantly increased risk (i.e., the 5–10% who meet criteria for familial or hereditary PDAC) should be identified to benefit from surveillance programs. PDAC screening is recommended when the risk of developing this neoplasm is significantly increased (>10-fold risk or >5% cumulative risk), such as FAMMM. Several recent studies analyzed the risk–benefit of PDAC surveillance in individuals at high risk, in which detection of PDAC during these programs resulted in a higher curative resection rate and a longer median survival time [16,17].

Until recently, it was considered that surveillance of *CDKN2A* GPV heterozygotes should only be performed if there was a family history of PDAC (CAPS 1–4 guideline [18]). However, according to the 2020 International Cancer of the Pancreas Screening Consortium (CAPS), PDAC surveillance is recommended in *CDKN2A* carriers regardless of family history of PDAC, starting at age 40 or 10 years before than the youngest relative affected [19]. These recommendations have changed in the last few years, since the most recent evidence seems to indicate that this syndrome carries a higher risk than previously reported [20]. Vasen et al. [21], and more recently (2022) Klatte et al. [22], published prospective studies in which PDAC surveillance seems effective in this group of individuals.

However, the yield of PDAC surveillance in *CDKN2A* GPV heterozygotes needs further studies to assess its efficacy in detecting pre-cancerous lesions, and improving the prognosis of PDAC through early detection. Our study aims at describing the performance of PDAC surveillance in the diagnosis of preneoplastic lesions or early-stages PDAC in *CDKN2A* GPV carriers.

## 2. Materials and Methods

### 2.1. Study Population

This is an observational analysis of prospectively collected data in a tertiary hospital. All *CDKN2A* germline pathogenic variant heterozygotes that were screened for PDAC and under follow-up at the high-risk digestive cancer clinic of th “Hospital Clínic de Barcelona” between June 2013 and July 2021 were included. Clinical criteria for genetic testing (*CDKN2A*, *CDK4* and *BAP1*) were: (a) patients diagnosed with at least two primary melanomas or with dual diagnosis of melanoma and PDAC, (b) families with at least one member with melanoma and two or more first- or second-degree relatives diagnosed with PDAC, (c) malignant melanoma in >1 first-degree relative, (d) first-degree relatives of *CDKN2A* GPV heterozygotes, and (e) presence of >50 nevi or multiple nevi with atypical histology.

All index cases were tested for all three genes (*CDKN2A*, *CDK4,* and *BAP1*), which are associated with a predisposition to melanoma, unless a *CDKN2A* pathogenic variant was known to exist in a first-degree relative. In that case, only *CDKN2A* was studied.

### 2.2. Definition of Surveillance Program

The included individuals were screened with annual magnetic resonance imaging (MRI) or endoscopic ultrasound (EUS) depending on availability or patient preferences, starting at age 50 or 10 years before the youngest affected relative, according to the European Society of Digestive Oncology (ESDO) expert discussion recommendations [23].

### 2.3. Data Recording

Personal and family history, environmental risk factors, and surveillance findings were analyzed.

-Personal data such as age, gender, alcohol consumption (current consumption of >14 units/week; former drinkers (>10 years) were not considered), smoking habits (current or former, with at least 5 pack-years of smoking), and comorbidities were included, as well as oncologic personal history. If PDAC: location, histology, treatment performed (surgery, radiotherapy, chemotherapy), “TNM” stage, survival, recurrence, cause of death;-Molecular data: genetic study (description of pathogenic variant);-Family history of cancer (digestive and extra-digestive neoplasms, type of cancer, age and degree of relationship);-PDAC surveillance: type of test (EUS or MRI), periodicity, findings (normal, preneoplastic lesions, cancer);-Definition of pancreatic lesions:

(a)High-risk pancreatic lesions [24]:

a.1. Any high-grade PanIN (III);

a.2. Any intraductal papillary mucinous neoplasia (IPMN) or mucinous cystic neoplasm with high-risk criteria, defined as: jaundice, presence of solid component, main pancreatic duct dilatation ≥10 mm, cyst diameter ≥4 cm, enhancing mural nodule >5 mm, thickened cyst wall, or abrupt change in pancreatic duct caliber with distal parenchymal atrophy.

(b)PDAC(c)Others (nonspecific lesions): IPMN o mucinous cystic neoplasm without high-risk criteria, nonspecific cyst, main pancreatic duct dilatation ≥10 mm, neuroendocrine tumor.

Side-branch duct IPMN and nonspecific cysts smaller than 10 mm were not considered as significant premalignant lesions.

### 2.4. Statistical Methods for Data Analysis

A descriptive analysis was performed where continuous quantitative variables were expressed with the median and interquartile range and quantitative variables with proportions.

The differences between qualitative variables were compared using Fisher’s test. The quantitative variables were analyzed using a non-parametric test (Mann–Whitney or Kruskal–Wallis for unpaired data and Wilcoxon for paired data). The association of PDAC or preneoplastic lesions and the variables of interest under study were carried out by means of a multivariate logistic regression analysis. A *p*-value less than 0.05 was considered statistically significant. For statistical analysis we used the SPSS 23 version (SPSS Inc., Chicago, IL, USA, 2021).

## 3. Results

### 3.1. General Characteristics

A total of 84 individuals with a proven *CDKN2A* germline pathogenic variant attended during the study period. After excluding 3 individuals due to age and 3 because they did not want to undergo surveillance, 78 individuals were finally included in the pancreatic surveillance program (see Figure 1).

Of the total cohort, 46 (59.0%) were women, with a median age at first test of 53 years, interquartile range (IQR) 47–63. Regarding environmental risk factors, the most prevalent was tobacco, with 31 (39.7%) individuals with a history of smoking (former or current smoker). A total of 17 patients (21.8%) had a family history of PDAC and 56 (71.8%) a personal history of melanoma (Table 1).

### 3.2. Pancreatic Cancer Surveillance Program: Characteristics and Findings

The median number of examinations per individual was three (IQR 1–4), with a total of 121 EUS and 116 MRI performed and a median follow-up of 66 months (IQR 42–81). Although the recommendation was annual surveillance, a group of patients explicitly requested to be tested biannually during the first years (at the beginning of the program in 2013).

In 50 (64.1%) individuals, the first test performed was MRI and in 28 (35.9%) it was EUS. In 36 (46.2%) individuals, both MRI and EUS were performed, in 17 (21.8%) only MRI, and in 25 (32.1%) only EUS.

A total of 17 pancreatic findings were detected in 16 (20.5%) individuals under surveillance. As can be seen in Figure 2, we identified 1 PDAC and 16 other pancreatic lesions considered low risk or uncertain meaning (nonspecific lesions). No significant premalignant lesions were identified, and a 61 (78.2%) individuals had a normal surveillance test.

Regarding the individual with PDAC, he was 58 years old when the first test was performed and was included for presenting a GPV in *CDKN2A* [exon 2, c.358delG (p.Glu120Serfs*26)]. He had family and personal history of melanoma. He had been diagnosed with a melanoma in situ 15 years ago, surgically treated and in which no systemic therapy was needed. He was a former smoker, without personal history of alcohol consumption or family history of PDAC. This patient was one of the seven individuals in our cohort who had a personal history of type 2 diabetes mellitus (T2DM) that was diagnosed in 2015, 4 years before the PDAC diagnosis. The neoplasm was located in the pancreatic tail and diagnosed on the first surveillance test by EUS and on an early stage (T2N0M0) (Figure 3). A distal pancreatectomy with splenectomy was performed after neoadjuvant chemotherapy. Fifteen months later it recurred with a single liver metastasis, and he is currently (33 months after the first surgery) alive undergoing chemotherapy treatment.

Regarding the other findings, all seven IPMN detected were small (<10 mm) and side-branch IPMN without any high-risk criteria. Of the three neuroendocrine tumors (NET), two were well-differentiated tumors, confirmed by EUS-guided fine needle aspiration with low (<2%) Ki-67 index (G1). In the third case, there was not enough histological sample and it was classified as a NET by the radiological characteristics. They presented a maximum diameter of 5, 16, and 7 mm, respectively; all three were considered low-risk lesions. The five nonspecific cysts were all <10 mm and no other diagnostic tests were performed. Regarding the main pancreatic duct dilatation, it was a slight nonspecific dilation (5 mm) detected by EUS, with a normal previous MRI. In these last scenarios, a wait-and-watch strategy was adopted. All lesions remained stable and did not show any warning sign during follow-up.

In our cohort, 10/17 (58.8%) lesions were detected by EUS, 1/17 (5.9%) by MRI, and in 6/17 (35.3%) by both tests. Of the 10 individuals with lesions diagnosed by EUS, 4 (40.0%) had a normal MRI and in the other 6 (60.0%), MRI was not performed. These four lesions were one main pancreatic duct dilatation, one intraductal papillary mucinous neoplasm, one neuroendocrine tumor, and one nonspecific cyst. In the case where the lesion was detected only by MRI, EUS had not been performed. The PDAC was detected on the first surveillance by EUS (see Table 2).

### 3.3. Risk Factors Associated with PDAC and Preneoplastic Lesions

The risk factors ‘smoking’, ‘alcohol consumption’, ‘Diabetes Mellitus’, or ‘family history of PDAC’ were not independently associated with an abnormal surveillance exam (*p*> 0.05) in the logistic regression analysis.

## 4. Discussion

PDAC screening has not been proven to prevent PDAC or reduce mortality, and the target population for PDAC surveillance remains uncertain. The most recent evidence suggests that individuals with germline pathogenic variants in *CDKN2A* have a significantly higher risk than others, similar to that of Peutz–Jeghers syndrome. A systematic review of 16 studies published in 2015 analyzed the risk–benefit of PDAC surveillance in high-risk individuals for this neoplasia. Detection of PDAC during surveillance resulted in a higher curative resection rate (60% vs. 25%, *p*-value = 0.011) and a longer median survival time (14.5 vs. 4 months, *p* < 0.001) compared to the control group [16]. Canto et al. observed in a 16 years follow-up study of 354 individuals at high risk of PDAC that the majority (90%) of cancers detected during surveillance were resectable with a 85% survival at 3 years [17]. This indirect evidence supports the performance of surveillance in patients with family history or proven pathogenic germline variants and suggests an increased survival in selected patients. MRI or preferably EUS (given its ability to take ultrasound-guided fine-needle aspiration and to detect lesions smaller than 10 mm) can diagnose precursor lesions and early cancer, giving these patients the possibility of undergoing curative treatments [25].

In our cohort of *CDKN2A* carriers, an early PDAC was detected and a potential curative treatment was offered to the patient, who after almost 3 years, is still alive. However, in up to 20% of the individuals, a pancreatic alteration was detected during surveillance without a malignant significance. The ability to detect small lesions in the pancreas, including any kind of NET, has grown dramatically in the last two decades [26], and this is, in part, due to the more frequent use of MRI and EUS in clinical practice. Thus, our benign surveillance findings seem to be similar to those expected in the general population older than 50 years [27,28] However, although the diagnostic yield of our pancreatic surveillance program is low, it seems to have an impact on survival. Our results are in consonance with a recent study [20] where FAMMM and Peutz–Jeghers syndrome seemed to have an increased risk of PDAC over other genetic syndromes. Overbeek et al. showed the yield of long-term surveillance in one of the largest cohorts of high-risk individuals to date (*n* = 366). PDAC was diagnosed in 10 of the 165 GPV heterozygotes, with a cumulative 10 year incidence of 9.3%, whereas in high-risk individuals without a GPV, there was not any case of PDAC (*n* = 201).

No environmental or family risk factors were independently associated with PDAC or other findings. It is worth noting that four individuals presented a lesion detected by EUS with a previous normal MRI. This study was not designed to identify differences between these two tests, but we must take into account other studies that have proven EUS as more sensitive in detecting small pancreatic lesions (parenchymal abnormalities, cysts, and ductal adenocarcinomas) [25].

Vasen et al. [21] published in 2016 a prospective follow-up study from three European expert centers including 178 *CDKN2A* GPV heterozygotes with a mean follow-up time of 53 months. They detected a PDAC in 13 (7.3%) of them, the resection rate was 75%, and the 5 year survival rate was 24%. More recently (2022), Klatte et al. [22] published a 20 year prospective follow-up study including 347 *CDKN2A* individuals of a Dutch hospital, in which a high incidence of PDAC (*n* = 36) was demonstrated with 5.6 years of follow-up, and most of them (*n* = 31) were resectable. Our study is based on a cohort from a low-incidence region of PDAC and this could justify the fact that our incidence of PDAC is lower than in these other series.

Overbeek et al. [29] described in a large cohort (more 2000) under surveillance in which more than half of high-risk individuals developing high-grade dysplasia or PDAC had no previous lesions detected, which suggests a low prevalence of these lesions can explain why no precursor lesions were detected in our cohort. Moreover, PanIN lesions require a better characterization for an adequate diagnosis, and for this reason, we consider that more personalized surveillance programs are necessary. In this sense, the use of biomarkers, such as dysregulated expression of miR-21, miR-155, miR-196, and miR-210 that have been observed in PanIN and IPMN [30], could be used to identify target populations, and prediction models for PDAC based on the interaction of modifiable and genetic risk factors are being developed and will help to identify those patients at a higher risk [31].

Our study has some limitations. First, as it is a surveillance study, the follow-up time is possibly insufficient to assess the expected lesions (median of examinations per individual of three, IQR 1–4) requiring longer follow-up to ensure the benefits of surveillance. On the other hand, the size of the cohort is possibly insufficient to evaluate lesions of such a low incidence as PDAC. However, one of the main strengths of our study is that it is one of the largest cohorts of *CDKN2A* GPV carriers and it is a homogenous cohort (only *CDKN2A*), compared to previous reports where patients with several types of hereditary syndromes and familial PDAC, with different risk of PDAC were assessed.

## 5. Conclusions

In our pancreatic cancer surveillance program, no significant high-risk precursor lesions were found, but one early PDAC was diagnosed. These results suggest that while better preventive strategies are developed, annual surveillance with EUS and/or MRI in *CDKN2A* carriers may be beneficial.

## Figures and Tables

**Figure 1 cancers-15-01690-f001:**
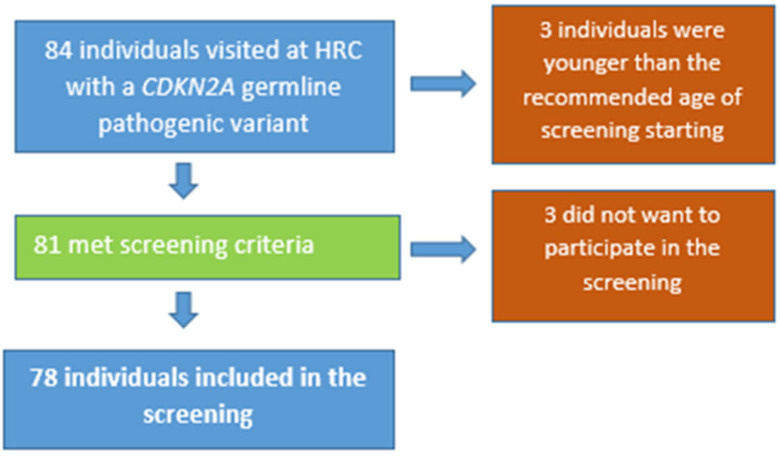
Flow chart—patients included in the study. HRC, high-risk clinic.

**Figure 2 cancers-15-01690-f002:**
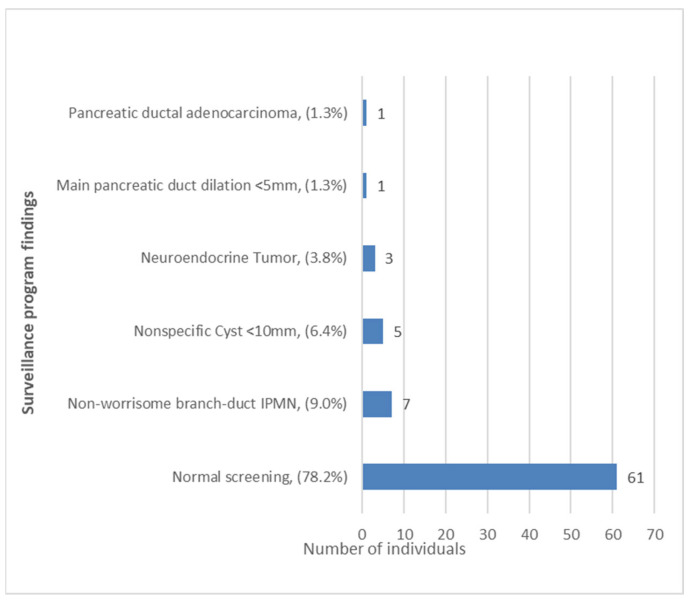
Surveillance findings. IPMN, intraductal papillary mucinous neoplasm.

**Figure 3 cancers-15-01690-f003:**
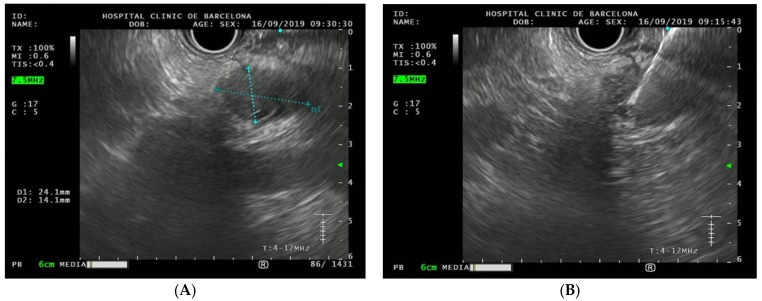
(**A**) Endoscopic ultrasound picture of the detected pancreatic cancer (24.1 × 14.1 mm). (**B**) Fine needle puncture in this operable tumor was performed and the cytology was positive for malignancy (adenocarcinoma). The final size was 28 × 16 mm (in the surgically removed piece).

**Table 1 cancers-15-01690-t001:** Cohort baseline characteristics (N = 78). IQR, interquartile range; PDAC, pancreatic ductal adenocarcinoma.

Characteristics of the Individuals Included (N = 78)	
Median age (years) at first test (median, IQR)	53 (47–63)
Female, n (%)	46 (59.0%)
Smoking habit (former or current smoker), n (%)	31 (39.7%)
Alcohol consumption, n (%)	5 (6.4%)
Diabetes mellitus type 2, n (%)	7 (9.0%)
Family history of PDAC, n (%) - First-degree relatives - Second and/or third-degree relatives	17 (21.8%) 5 (6.4%) 12 (15.4%)
Personal history of cancer - Melanoma - Prostate - Breast - Ovary - Brain	56 (71.8%) 2 (2.6%) 2 (2.6%) 1 (1.3%) 1 (1.3%)

**Table 2 cancers-15-01690-t002:** Surveillance findings, *n* = 16 in 17 individuals. EUS, endoscopic ultrasound; MRI, magnetic resonance imaging; IPMN, intraductal papillary mucinous neoplasm. EUS or MRI was performed during follow-up, depending on availability, patient preference, and previous test findings.

Lesions Detected at the Surveillance Program (*n* = 17)	Test Performed During Follow-Up		Test that Detected the Lesion	
	EUS	MRI	EUS	MRI
Main pancreatic duct dilation (5 mm)	Yes	Yes	Yes	No
IPMN	Yes	Yes	Yes	Yes
IPMN	Yes	Yes	Yes	Yes
IPMN	No	Yes	Not performed	Yes
IPMN	Yes	No	Yes	Not performed
IPMN	Yes	No	Yes	Not performed
IPMN	Yes	Yes	Yes	Yes
IPMN	Yes	Yes	Yes	No
Neuroendocrine tumor	Yes	Yes	Yes	No
Neuroendocrine tumor	Yes	Yes	Yes	Yes
Neuroendocrine tumor	Yes	Yes	Yes	Yes
Pancreatic cancer	Yes	No	Yes	Not performed
Nonspecific cyst	Yes	Yes	Yes	No
Nonspecific cyst	Yes	No	Yes	Not performed
Nonspecific cyst	Yes	Yes	Yes	Yes
Nonspecific cyst	Yes	No	Yes	Not performed
Nonspecific cyst	Yes	No	Yes	Not performed

## Data Availability

The data that support the findings of this study are available from the corresponding author upon reasonable request.

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
