# Peer review of "Pancreatic Cancer Surveillance in Carriers of a Germline Pathogenic Variant in CDKN2A"

_cancers, 2023, doi:10.3390/cancers15061690_

Round 1

Reviewer 1 Report

The authors present a well written report(prospective study) on screening of pancreatic cancer in CDKN2A mutation carriers. Their study showed that screening of PDAC in high risk population makes sense( one detected cancer, one saved or at least improved survival).  If I could change something and make the study more interesting to the readers I would add the endosonographic picture of detected cancer and provide the information about the EUS -FNA/B( was the biopsy in this operable tumor performed or the diagnosis was made judging on eur features?What was the exact size of the detected cancer?

I would also provide and the underline  the information about the  neuroendocrine tumors (size matters). If the were more than 2cm in size, I would classify them as high risk lesions. Isin't it interesting that screening detected three NET tumors? Does the CDKN2A mutation increase the risk of neuroendocrine pancreatic tumors? What does literature say about that? Maybe, this aspect is worth of underlining it in discussion.  

Reviewer 2 Report

Dr. Llach et al detail a retrospective on CKDN2A GPV carriers. This is a valuable study with a large sample size for a rare population. My concerns are as follows. 

1.       Were any of these 78 patients tested for other germline or somatic pathological variants? If so, details would be helpful to the reader.

2.       Patients were screened “annually”, but only received a median of 3 examinations during a median follow-up period of 66 months (5.5 years).Were they really screened annually or, for example, biannually in some cases at the discretion of the physician? Was there an annual screening protocol in place?

3.       In Methods 2.1, the authors state the clinical criteria for CDKN2A genetic testing. It would be interesting to show what precent of screened patients in each of the 5 criteria were positive for CDKN2A GPV.

4.       The authors state that they followed ESDO expert discussion recommendations (line 113). However, Reference 23 does not recommend annual screening for patients with FAMMM. Please rephrase Methods 2.2.

5.       Line 137: “main pancreatic duct dilatation ≥10mm” and Line 142: “Wirsung dilatation < 10mm): I would use the same terminology for both.

6.       Table 1: Does “alcohol consumption” refer to former or current drinkers? How much per day? It is surprising that only 6.4% consume alcohol. Please provide a definition.

7.       The limitations paragraph should come at the end of the Discussion section.

8.       The authors conclude that “These results support previous evidence in favor of recommending surveillance in CDKN2A GPV heterozygotes” based on one PDAC being discovered out of 78 at-risk patients. I would weaken this statement.

9.       EUS/MRI is costly. Was abdominal ultrasound performed in these patients? If so, how did abdominal ultrasound fare in finding pancreatic lesions relative to EUS/MRI (especially in the case with pancreatic cancer)?

Round 2

Reviewer 2 Report

The authors have adequately revised their manuscript. I have no further comments.